# The Impact of Non-Economic Factors on Voluntary Tax Compliance Behavior: A Case Study of Small and Medium Enterprises in Vietnam

Thu Hien Nguyen

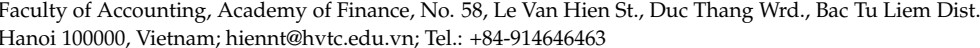

Faculty of Accounting, Academy of Finance, No. 58, Le Van Hien St., Duc Thang Wrd., Bac Tu Liem Dist., Hanoi 100000, Vietnam; hiennt@hvtc.edu.vn; Tel.: +84-914646463

**Abstract:** One of the main reasons governments of developing countries, including Vietnam, face many difficulties in tax collection is taxpayers' non-compliance with taxes. Therefore, the question for the governments of these countries is how to encourage taxpayers to comply voluntarily without having to resort to mandatory measures. This study examines non-economic factors affecting the voluntary tax compliance behavior of small and medium enterprises (SMEs) in Vietnam. The data used in the study were collected from the survey results of 339 tax accountants and managers at SMEs in some provinces and cities in Vietnam. The study data were processed with the statistical software SPSS 20 to test for Cronbach's alpha, exploratory factor analysis, and linear regression analysis. The research results show that the possibility of tax inspection and audit, social norms, tax knowledge, personal norms, perception of the tax system's fairness, and tax service quality has a significant influence on the voluntary tax compliance behavior of SMEs in Vietnam; of which, the possibility of tax inspection and audit has the strongest impact, and tax services quality has the weakest effect on the voluntary tax compliance behavior of these enterprises.

**Keywords:** voluntary tax compliance; taxpayers; SMEs; tax inspection and audit; social norms; tax knowledge; personal norms; tax system's fairness; tax service

## 1. Introduction

Taxes are one of the crucial factors in managing national income, especially in developed countries, and taxes have played a central role in societies since their inception thousands of years ago (Lymer and Oats 2009). Taxes are an essential tool for governments to regulate the macroeconomy, promote investment, control inflation, and redistribute wealth and income in society. Today the role of government in countries is increasing, and they have to collect more taxes to finance their activities. Even so, governments are finding it difficult to collect the taxes they need for various reasons. One of the main reasons that must be mentioned is taxpayers' non-compliance. Tax compliance is the extent to which taxpayers comply with their tax obligations as set out in tax law (James and Alley 1999); taxpayers file tax returns appropriately and fulfill their tax obligations according to regulations tax law (Hamm 1995). Voluntary tax compliance is the correct, complete, and timely payment of taxes without the need for coercive efforts by the government (Kirchler et al. 2008). For the voluntary system to work effectively, taxpayers must trust that taxes are levied relatively and that everyone pays their share. They will voluntarily comply because they feel obligated to do so as community members (Kirchler et al. 2008). The fact shows that, similar to some other developing countries, taxpayers' non-compliance is always a challenge for the government and policymakers of Vietnam because it is subject to influence by many different factors. In Vietnam, tax violations have been common in almost all taxes in recent years. Tax fraud is becoming more complicated; the scope and the scale is getting bigger, and the tricks are getting more sophisticated. To overcome this problem, the Vietnamese government has introduced many measures to limit tax non-compliance as well

as improve people's understanding of tax laws, awareness of their rights, and obligations and responsibilities, thereby helping the government improve public revenues to perform management functions for socio-economic stability and development. To ensure financial and budget resources to contribute to the implementation of socio-economic development goals and tasks for the years 2021–2030, the Government of Vietnam has approved a financial strategy for 2030, according to which the rate of mobilization to the state budget from taxes and fees in the 2021–2025 period is about 13–14% of the GDP and about 14–15% of the GDP in the 2026–2030 period (The Prime Minister 2022). However, with the peculiarity of being a developing country, in the past, Vietnam's tax system has regularly undergone legal amendments and mainly focused on law enforcement as a remedy to make sure the tax system works appropriately. Therefore, many taxpayers have not supported voluntary tax compliance, and they seem to be facing specific difficulties (for example, tax knowledge is limited, they have to update too many changes due to legal changes, and tax authorities still use administrative orders to impose unfair treatment on taxpayers, etc.). This can be considered one of the difficulties and challenges of the internal weaknesses of mechanisms and policies that the Vietnamese government has not thoroughly resolved. Therefore, the question for the government and the planners is how to make taxpayers comply voluntarily without using mandatory measures. To solve this problem, the Vietnamese government needs to perfect the tax administration institution to ensure synchronization, publicity, transparency, and fairness, apply risk management in tax administration and create favorable conditions for voluntary compliance by taxpayers. At the same time, concretize action plans through reforming administrative procedures to increase satisfaction, create taxpayers' confidence in the government, and improve the tax morale of the people. On the other hand, in recent years, many studies have analyzed factors affecting taxpayers' tax compliance behavior. Some studies have considered non-economic factors such as psychological, ethical, and social factors affecting the tax compliance behavior of taxpayers. Even so, the results of the studies still have some disparities and are inconsistent. Furthermore, previous studies on tax compliance and voluntary tax compliance were mainly conducted in developed countries with synchronous infrastructure and a fairly complete tax law system. Research on tax compliance, especially voluntary tax compliance in countries with economies in transition such as in Vietnam, is still quite limited. Therefore, to help carry out the above efforts, this study examines non-economic factors affecting taxpayers' voluntary tax compliance behavior in small and medium enterprises (SMEs), that is, which factors motivate taxpayers to comply with tax laws compliance with the tax system or what factors influence taxpayers' non-compliance with the tax system. This study shows that all six factors included in the research model affect SMEs' voluntary tax compliance behavior, including the possibility of tax inspection and audit, social norms, tax knowledge, personal norms, perception of the tax system's fairness, and tax service quality. The findings of this study provide evidence to help the government, policymakers, and tax authorities in Vietnam combine measures to solve these difficulties in system development or enact tax policies that promote voluntary tax compliance by taxpayers.

## 2. Literature Review

Compliance behavior has been applied by Becker (1968) in the study of crime and punishment: an economic approach. In his approach, Becker assumed that tax compliance behavior is determined by the monetary benefits received. Individuals will commit an offense when their expected benefits outweigh the possible benefits of doing other things through lawful activities within a certain period. Taxpayers will evade taxes if the consequences of being caught and punished are less than the amount evaded. Then, Allingham and Sandmo (1972) extended and developed a formal model for analyzing tax evasion (economic deterrence model). The economic deterrence model is believed to be the first to study tax compliance and examine the effectiveness of deterrence through sanctions for illegal behavior, such as taxpayers' non-compliance is based on choice under risk and uncertainty. Allingham and Sandmo (1972) assume that taxpayers are utility maximizers with a realistic

knowledge of the penalty and the likelihood of detection. The conventional deterrent model emphasizes sanctions as a critical determinant in combating tax non-compliance issues. However, introducing stringent punishment will not be practical or effective as a deterrent if the violator knows that the possibility of detection is very high. Deterrence is approached in punitive and persuasive ways (Fischer et al. 1992). In which sanctions focus on stricter and harsher penalties to punish non-compliance; method of persuasion leaning toward moral values. The persuasion measure assumes that compliance will be achieved when tax authorities attract the tax morale of taxpayers. There are mechanisms and policies to encourage, support, and create favorable conditions for taxpayers to comply voluntarily without having to be deterred.

Tax compliance is a fairly broad topic and is considered by many researchers, so the definition of tax compliance is determined in different ways, depending on the nature and object of study. Tax compliance can be understood as the accurate declaration of income and expenses by tax law provisions (Alm et al. 1991), ensuring on time without the authorities' tax managers (Jackson and Milliron 1986). Tax compliance reflects a taxpayer's willingness to pay taxes (Kirchler 2007) under tax law to achieve a country's economic equilibrium (Andreoni et al. 1998). However, a taxpayer's ability and willingness to comply with tax laws is determined by the ethics, regulatory environment, and other situational factors at a particular time and place (Song and Yarbrough 1978). Taxpayers will make a tax compliance decision by weighing the risks encountered with the expectation of benefits from unreported income (Allingham and Sandmo 1972). Voluntary intention to comply is described as an interaction between a taxpayer's trust in the government and the authority's right to monitor taxpayers. When trust in the authorities is high, taxpayers will have a voluntary intention to pay taxes. Voluntary compliance stems from a taxpayer's willingness to cooperate skillfully and a moral obligation to contribute to the public good. Taxpayers consider it an obligation as a citizen even if tax audits do not exist; they are sure they are doing the right thing even though others are not doing it; they want to help support the state and other citizens as well as contribute to the good of all (Kirchler and Wahl 2010), so voluntary tax compliance is primarily determined by the taxpayer's trust in the tax authorities (Kirchler et al. 2008). Taxpayers voluntarily pay taxes even under low tax enforcement because of their intrinsic motivation, stemming from the perception that paying taxes is an obligation (Cummings et al. 2009), which shows that tax morale has a positive influence on tax compliance behavior (Alabede et al. 2011; Richardson 2006). The term "tax morale" has been used by Luttmer and Singhal (2014) as a shorthand for any such nonpecuniary factors as well as deviations from expected utility maximization. Types of intrinsic motivation can cause people to comply with laws and expectations. Other forms of intrinsic motivation are feelings of pride and a positive self-image often associated with honesty, civic duty, and altruism toward others. Individuals may feel guilt or shame for not complying. In addition, they may comply due to reciprocal motives: the willingness to pay taxes in exchange for benefits that the state provides to them or others even though their monetary rewards will be higher if they do not pay taxes; their willingness to pay taxes depends on the individual's relationship to the state and not on the direct relationship of the tax benefit, or they may be influenced by peer behavior, and the possibility of social recognition or punishment from peers, cultural or social norms may influence the strength of motivations intrinsic, reciprocal motivation or sensitivity to peers. This shows that tax morale plays a significant role in tax compliance behavior. In another aspect, Doerrenberg and Peichl (2018) add a reciprocity component by reminding participants that tax compliance and government services are closely linked, and reciprocity increases tax morale more than standard social treatment. In addition, the authors found that risk aversion is positively correlated with tax morale. In a study in a developing country that belongs to the Association of South East Asian Nations and has some conditions similar to Vietnam, Taing and Chang (2020) confirm that a positive attitude towards tax (tax morale) significantly affects the compliance intention of Phnom Penh residents, Cambodia. Positive tax morale is crucial to improving the tax compliance of taxpayers.

## 3. Research Hypothesis and Model

### 3.1. The Possibility of Tax Inspection and Audit

Tax inspection aims to assess the completeness and accuracy of information and documents in tax records or assess taxpayers' compliance with tax laws. Tax audits positively reduce tax fraud (Jackson and Jaouen 1989), contributing to the early prevention of tax evasion attempts and forcing taxpayers to comply with taxes. In addition, it appears that the fear of being audited has a stronger impact on tax compliance than the potential penalty on evaded tax (Muehlbacher and Kirchler 2010; Kastlunger et al. 2013); an increased likelihood of tax audits leads to an increase in tax compliance (Kirchler 2007; Inasius 2018). In a tax system operated by self-declaration, self-calculation, and self-payment, the possibility of tax inspection plays an indispensable role in enhancing and positively impacting the voluntary tax compliance of taxpayers. In other words, audit probability directly affects voluntary tax compliance. Thorough tax inspection and audit can encourage taxpayers to be more cautious in tax compliance. Taxpayers are aware of the adverse consequences and severe penalties, possibly even criminal penalties, if they are found to be non-compliant through tax audits and inspections. This has made taxpayers' sense of tax compliance better. Based on the above arguments, the first hypothesis of the study is stated as follows:

**Hypothesis 1 (H1).** *The possibility of tax inspection and audit positively influences taxpayers' voluntary tax compliance behavior.*

### 3.2. Social Norms

Ethical norms have a strong relationship with tax compliance (Battiston and Gamba 2013; Traxler 2010) and affect tax compliance behavioral intentions (Bobek et al. 2007); thus, social pressure has a strong positive effect on taxpayers' voluntary tax compliance behavior (Battiston and Gamba 2016). This is because taxpayers in a social community will form their intention to comply with taxes because of social acceptance (Benk et al. 2011), perceptions as well as invisible pressure from the public that society will influence their tax compliance or non-compliance decisions (Alm and McKee 1998; Torgler 2007; Jackson and Milliron 1986). For example, if taxpayers perceive non-compliance as common behavior among people around them, they may also choose not to comply with taxes (Kirchler et al. 2008). Social norms affect voluntary and mandatory tax compliance (Liu 2014). Therefore, a taxpayer's willingness to comply with taxes depends on the characteristics of the economic, political, and cultural environment at a particular time and place. Social norms are raised, which means that members of society will be aware of their roles and responsibilities towards paying taxes to ensure the country's economic development and social security. Meanwhile, taxpayers may be influenced by the social community so that the social acceptance attitude will affect their tax compliance decision; they will become more self-aware in compliance with tax. Through the above arguments, the second hypothesis of the study is stated as follows:

**Hypothesis 2 (H2).** *Social norms have a significant and positive relationship with taxpayers' voluntary tax compliance behavior.*

### 3.3. Tax Knowledge

Many different studies have examined the effect of tax knowledge on tax compliance behavior. Tax knowledge has a very close relationship with taxpayers' ability to understand tax laws and compliance (Singh and Bhupalan 2001). Eriksen and Fallan (1996) argue that the education level of taxpayers is an important factor affecting their general understanding of taxes, especially tax laws and regulations. More about taxes can improve tax attitudes towards increasing compliance and reducing the propensity to tax evasion. In contrast, low tax knowledge has a negative impact on taxes (Lewis 1982), and tax knowledge is the strongest predictor of tax compliance (Inasius 2015). In order to raise social awareness about tax compliance, mass communication programs need to be promoted. In addition, legal

education needs to be included in educational curricula at all levels with appropriate forms, which will create a premise for voluntary tax compliance. Taxpayers' knowledge of tax laws will help them perform the registration and declaration procedures well and determine the amount of tax payable and the adverse consequences of tax fraud. Thus, these insights will improve the compliance attitude of taxpayers. Through the above arguments, the third hypothesis of the study is stated as follows:

**Hypothesis 3 (H3).** *Tax knowledge has a direct and positive influence on taxpayers' voluntary tax compliance behavior.*

*3.4. Personal Norms*

Personal norms are the moral norms of individuals and the expectations of their behavior. Personal norms can develop by accumulating social norms that the individual perceives (Wenzel 2004). Some important social norms become the moral norms of the individual. Personal norms reflect individuals' beliefs, affecting their tax compliance behavior. Thus, tax ethics describe an individual's moral principles or values for paying taxes (Torgler and Murphy 2004). Tax ethics can also be seen as a moral obligation to pay taxes or a belief in contributing to society by paying taxes (Frey 1997). Taxpayers with good moral norms will tend to behave honestly and follow established rules, which impacts taxpayers' compliance with tax obligations (Wenzel 2004). Therefore, tax ethics are an important factor in reducing tax evasion and increasing voluntary tax compliance (Alm and Torgler 2006). Personal norms reflect their beliefs, affecting their tax compliance behavior. If taxpayers decline in morality, they will have deviant acts and support acts of taking advantage of loopholes in the law or the lax management of tax authorities to evade tax obligations. Apparently, they have not considered tax compliance behavior ethical behavior but only considered it a mandatory action according to the provisions of the law. Based on the above statements, the fourth hypothesis of the study is stated as follows:

**Hypothesis 4 (H4).** *Personal norms have a significant and positive relationship with taxpayers' voluntary tax compliance behavior.*

*3.5. Perception of the Tax System's Fairness*

Fairness is always a concern for people when performing a certain obligation; fairness is an objective requirement in the development of taxes. People often share and address concerns about the tax system's fairness (Rawlings 2003; Taylor 2003). Taxpayers' perception of the tax system's fairness affects their propensity to avoid or evade taxes (Jackson and Milliron 1986). Taxpayers will comply to fulfill their tax obligations if they find the tax system to be reasonably regulated. A fair tax system would increase the confidence of taxpayers, which in turn increases the willingness to comply with voluntary taxes. Conversely, taxpayers intend to evade taxes if they perceive the tax system as unfair (Vogel 1974). The degree of tax non-compliance is influenced by perceived procedural inequity, measured by operational inconsistency and lack of regulatory clarity (Kim and Lee 2020). There is a growing consensus in tax compliance studies that a fair tax system plays an important role in determining tax compliance (e.g., Amina and Saniya 2015; Inasius 2018; Taing and Chang 2020). The issue of transparency in spending on welfare, anti-corruption, and other social problems of the government will create a sense of fairness in taxpayers' perception. As a result, they feel their contributions are appropriately spent and benefit society and them personally, which in turn encourages taxpayers to comply voluntarily. Based on the foregoing, the fifth hypothesis of the study is stated as follows:

**Hypothesis 5 (H5).** *Perception of the tax system's fairness positively influences taxpayers' voluntary tax compliance behavior.*

*3.6. Tax Service Quality*

Tax is a typical type of public service provided by tax authorities. Tax service quality refers to the availability of services and facilities to serve taxpayers (Obid and Bojuwon 2014). The tax service quality includes the quality of tax support services, the inspection and examination team quality, the quality of the tax declaration-payment system, etc. In the modern tax management trend, taxpayers are seen as customers rather than perpetrators (Gangl et al. 2013); therefore, tax services are associated with high quality is an effective solution in improving taxpayers' satisfaction in tax administration. On the other hand, voluntary tax compliance is based on trust and the tax service environment. Therefore, strengthening tax services and improving organizational structure to support businesses will improve the tax compliance of taxpayers. Many studies have confirmed that the tax service quality has an influence on the tax compliance of taxpayers, in which the tax service quality is measured by criteria such as responsiveness, reliability, reassurance, empathy, and the facilities of the tax authorities (e.g., Obid and Bojuwon 2014; Hidayat et al. 2014). The fact that tax authorities create trust and satisfaction for taxpayers through the service environment will make taxpayers feel happy and ready to implement the recommendations of tax authorities and tax regulations of the state. This will incentivize taxpayers to switch to voluntary tax compliance without requiring mandatory measures. Based on the above arguments, the sixth hypothesis of the study is stated as follows:

**Hypothesis 6 (H6).** *Tax service quality has a significant and positive relationship with taxpayers' voluntary tax compliance behavior.*

*3.7. Demographic*

Many previous studies support a positive relationship between demographic characteristics such as age, gender, education, and occupation to tax compliance. Jackson and Milliron (1986) found that age, gender, and education positively correlate with taxpayer compliance. The authors imply that older taxpayers exhibit better tax compliance behavior; women are less likely to evade taxes than men, and educated taxpayers are more compliant than untrained taxpayers. Chung and Trivedi (2003) suggested that age positively correlates with compliance. Hasseldine and Hite (2003) also found that women pay more taxes than men. Chan et al. (2000) suggest that educational attainment is directly related to compliance. Trained taxpayers may be aware of non-compliance opportunities, but their potential better understanding of the tax system and a higher degree of ethical development will boost taxpayer attitudes to more favorable and compliant. However, because this study aims to examine non-economic variables affecting voluntary tax compliance, the author does not go into depth analysis of the influence of demographic characteristics. This study only considers the demographic factors integrated into the model from the perspective of controlled variables to better confirm the causal nature of non-economic variables. Therefore, the author does not build a hypothesis in this study but only uses age, gender, education, and occupation as the controlled demographic variables.

*3.8. Research Models*

Based on the above hypotheses, the author proposes the expected theoretical research model, as described in Figure 1.

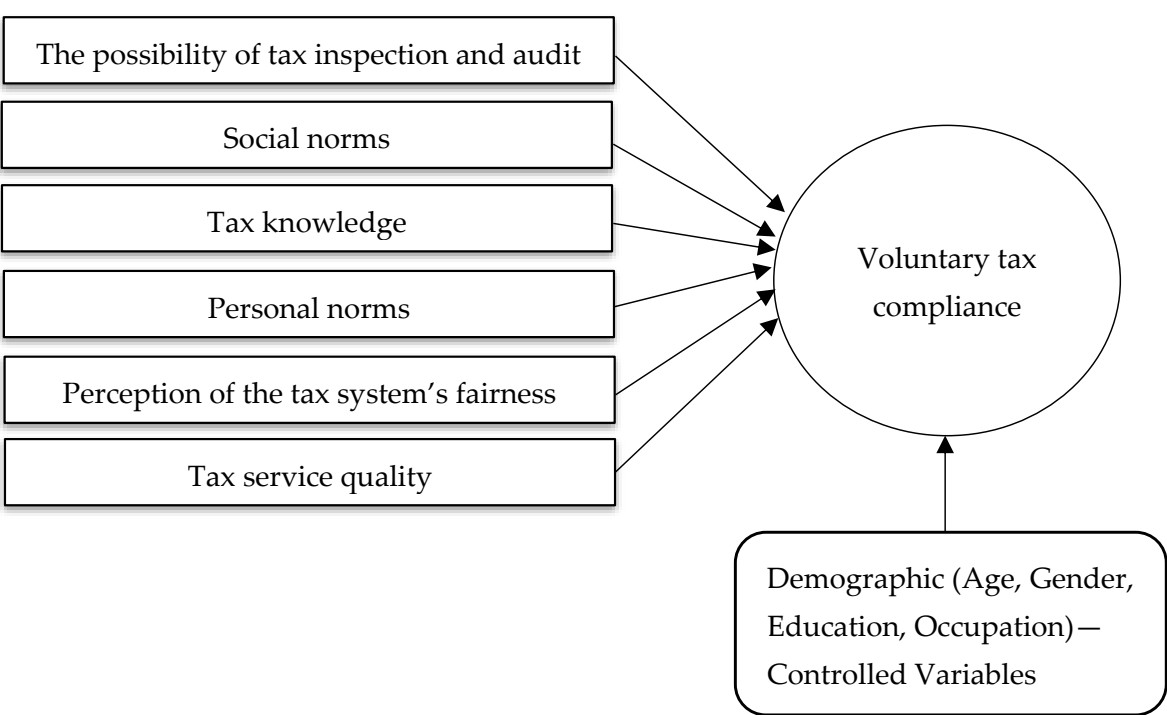

**Figure 1.** Proposed research model.

Based on the above hypotheses and theoretical research models, the author proposes a regression equation that reflects the relationship between "non-economic factors affecting small and medium enterprises' voluntary tax compliance behavior" is built as follows:

$$VTC_i = \alpha + \beta_1 AT_i + \beta_2 SN_i + \beta_3 KT_i + \beta_4 PN_i + \beta_5 FT_i + \beta_6 ST_i + \varepsilon_i \tag{1}$$

In which, VTC = Voluntary tax compliance behavior of SMEs, AT = Possibility of tax inspection and audit, SN = Social norms, KT = Tax knowledge, PN = Personal norms, FT = Perception of the tax system's fairness, and ST = Tax service quality.

$\alpha$: Constant term
$\beta_i$: Coefficients of the independent variables
$\varepsilon_i$: Residual

## 4. Research Methodology

### 4.1. Sampling Method

The sampling process mainly involves identifying the target audience, choosing a sampling method, determining the sample size, and selecting sample factors (Zikmund 2003). The sample size depends on the regression model processing method, necessary reliability, etc. Usually, in exploratory factor analysis (EFA), the sample size must be equal to 4 or 5 times the number of variables in factor analysis (Hoang and Ngoc 2008). In practical research applications, the sample size is often larger than 150 (Anderson and Gerbing 1988). According to Tabachnick and Fidell (2007), in multiple regression analysis, the formula commonly used to determine sample size is $n \geq 50 + 8p$ (where n is the minimum sample size; p is the number of independent variables in the model). The sample of this study was collected from SMEs in Vietnam. In Vietnam, SMEs are defined by many different criteria. In this study, the author identifies SMEs based on the criteria of the average number of employees participating in social insurance per year, which, for enterprises in the fields of agriculture, forestry, fisheries, industry, and construction employing no more than 200 people; enterprises in the field of commerce and services employing no more than 100 people (Government 2021). Therefore, this study selects a survey of SMEs, which is completely suitable for the actual conditions in Vietnam. Currently, Vietnam has more than

800,000 enterprises in operation, of which SMEs account for 97%, contributing 45% of GDP and 31% of the total budget (Sam 2021).

*4.2. Data Collection Methods*

The method of data collection through a questionnaire survey is a means to obtain more reliable survey responses with a possibility of achieving a higher response rate, thus improving the validity of this study. The data used in this study were collected through a survey of managers (directors, deputy directors, chief accountants) and tax accountants at 400 SMEs in the provinces and cities of Vietnam. These surveyed subjects are said to have a very good understanding of the internal control system, financial management system, or the business's current situation, so they have a very important and decisive role in corporate tax compliance. After removing invalid questionnaires, there remained 339 valid questionnaires (an 84.75 percent response rate). The survey period was from November 2021 to February 2022. In this study, the characteristics of the study sample are shown in Table 1.

**Table 1.** Description of the characteristics of the study sample (N = 339).

| Characteristics | Classification | Frequency | Percentage (%) | Code |
|---|---|---|---|---|
| Occupation | Director | 44 | 13 | 1 |
| | Deputy Director | 59 | 17 | 2 |
| | Chief accountant | 98 | 29 | 3 |
| | Tax accountant | 138 | 41 | 4 |
| Working time | Less than 5 years | 36 | 11 | - |
| | From 5 to less than 10 years | 58 | 17 | - |
| | From 10 to less than 20 years | 144 | 42 | - |
| | Over 20 years | 101 | 30 | - |
| Education | Postgraduate level | 45 | 13 | 1 |
| | University degree | 246 | 73 | 2 |
| | College degree | 48 | 14 | 3 |
| Age | From 18 to 25 years old | 34 | 10 | 1 |
| | From 25 to 35 years old | 108 | 32 | 2 |
| | From 35 to 50 years old | 110 | 32 | 3 |
| | Over 50 years old | 87 | 26 | 4 |
| Gender | Male | 201 | 59 | 1 |
| | Female | 138 | 41 | 0 |

The author conducted a survey of tax managers and accountants by a direct interview method or via email or post using a survey questionnaire with a 5-point Likert scale (Strongly disagree = 1, Disagree = 2, Neutral = 3, Agree = 4, Strongly agree = 5); the use of Likert scale was to make it easier for respondents to answer the questions in a simple way. To achieve this target, the present investigation built up a model which depended on prior research studies, and the model appears in Figure 1. Furthermore, the present examination used seven variables, as presented in the regression Equation (1). The seven variables used in this examination are the VTC, AT, SN, KT, PN, FT, and ST; of which, six items of VTC were adopted from the prior studies by Song and Yarbrough (1978), Jackson and Milliron (1986), and Kirchler and Wahl (2010). The three items of AT were adopted from the prior studies by Kirchler (2007) and Inasius (2018). Next, three items of SN were adopted from the earlier research by Kirchler et al. (2008), Benk et al. (2011), Liu (2014), and Battiston and Gamba (2016). The five items of KT were adopted from the prior studies by Singh and

Bhupalan (2001), Eriksen and Fallan (1996), and Inasius (2015). The three items of PN were adopted from the prior studies by Torgler and Murphy (2004), Wenzel (2004), and Alm and Torgler (2006). The six items of FT were adopted from the prior studies by Inasius (2018), Kim and Lee (2020), and Taing and Chang (2020). Finally, four items of ST were adopted from the previous studies by Gangl et al. (2013), Obid and Bojuwon (2014), and Hidayat et al. (2014). In addition, for demographic variables, the gender variable was coded as a dummy variable (Male = 1, Female = 0). Finally, the age, education, and occupation variables were coded as ordinal with their respective levels (Table 1).

### 4.3. Data Analysis Methods

The study used the statistical software SPSS 20 for descriptive statistical analysis through a multi-variable linear regression model with criteria such as Cronbach's Alpha test to assess the reliability of the scale, EFA analysis to regroup observed variables into more meaningful factors, and test linear regression models to determine the correlation between non-economic factors and the voluntary tax compliance behavior of SMEs in Vietnam.

## 5. Research Results

### 5.1. Cronbach's Alpha Test

Cronbach's Alpha coefficients examine the degree of correlation between observed variables in the same factor included in the research model. The results in Table 2 show that all of the variables have Cronbach's alpha coefficients greater than 0.6, so the scales can be used well and reliably (Hoang and Ngoc 2008). Thus, the Cronbach's Alpha test results show that the observed variables belonging to the factor groups remain the same.

**Table 2.** Cronbach's Alpha test.

| | Component | N of Items | Cronbach's Alpha |
|---|---|---|---|
| 1 | The possibility of tax inspection and audit (AT) | 3 | 0.826 |
| 2 | Social norms (SN) | 3 | 0.834 |
| 3 | Tax knowledge (KT) | 5 | 0.815 |
| 4 | Personal norms (PN) | 3 | 0.834 |
| 5 | Perception of the tax system's fairness (FT) | 6 | 0.882 |
| 6 | Tax service quality (ST) | 4 | 0.808 |
| 7 | Voluntary tax compliance (VTC) | 6 | 0.850 |

### 5.2. Exploratory Factor Analysis (EFA)

The exploratory factor analysis of the independent variables in Table 3 shows that the KMO coefficient is 0.744 (satisfying $0.5 \leq$ KMO $\leq 1$), which should be satisfactory. The Barlett's test has a Sig value of less than 5% ($p < 0.01$); therefore, these observed variables are closely correlated and suitable for EFA analysis. The total variance extracted is 67.224% (greater than 50%), and the Eigenvalues is 1.936 (greater than 1), so it is satisfactory (Anderson and Gerbing 1988). These observed variables all have factor loading coefficients greater than 0.5, so they are satisfactory (Hair et al. 1998). Thus, the results of the EFA analysis are completely consistent, and the extracted factors are reliable and valid.

**Table 3.** Exploratory factor analysis results of independent variables.

| | Component | | | | | |
|---|---|---|---|---|---|---|
| | **1** | **2** | **3** | **4** | **5** | **6** |
| FT5 | 0.886 | | | | | |
| FT4 | 0.844 | | | | | |
| FT2 | 0.790 | | | | | |
| FT6 | 0.748 | | | | | |
| FT1 | 0.719 | | | | | |
| FT3 | 0.714 | | | | | |
| KT4 | | 0.876 | | | | |
| KT2 | | 0.759 | | | | |
| KT3 | | 0.717 | | | | |
| KT5 | | 0.691 | | | | |
| KT1 | | 0.680 | | | | |
| ST1 | | | 0.826 | | | |
| ST4 | | | 0.786 | | | |
| ST2 | | | 0.785 | | | |
| ST3 | | | 0.780 | | | |
| SN2 | | | | 0.904 | | |
| SN1 | | | | 0.868 | | |
| SN3 | | | | 0.833 | | |
| PN2 | | | | | 0.878 | |
| PN3 | | | | | 0.865 | |
| PN1 | | | | | 0.837 | |
| AT2 | | | | | | 0.938 |
| AT3 | | | | | | 0.857 |
| AT1 | | | | | | 0.780 |

| Kaiser–Meyer–Olkin Measure of Sampling Adequacy. | KMO | 0.744 |
|---|---|---|
| Bartlett's Test of Sphericity | Sig. | 0.000 |
| Extraction Sums of Squared Loadings | Total | 1.936 |
| | Cumulative % | 67.224 |

The results of the exploratory factor analysis of the dependent variable in Table 4 show that the coefficient KMO is 0.852 (satisfying $0.5 \leq KMO \leq 1$), and the Barlett's test has a Sig value of less than 5% ($p < 0.01$), so the model is suitable for analysis; the variables are correlated in the population. The total variance extracted is 57.802% (greater than 50%); Eigenvalues is 3.468 (greater than 1), and the model is eligible for EFA analysis.

**Table 4.** Exploratory factor analysis results of the dependent variable.

| | Component |
|---|---|
| | **1** |
| VTC6 | 0.843 |
| VTC5 | 0.808 |
| VTC3 | 0.797 |
| VTC1 | 0.748 |
| VTC4 | 0.683 |
| VTC2 | 0.666 |

| Kaiser-Meyer-Olkin Measure of Sampling Adequacy. | KMO | 0.852 |
|---|---|---|
| Bartlett's Test of Sphericity | Sig. | 0.000 |
| Extraction Sums of Squared Loadings | Total | 3.468 |
| | Cumulative % | 57.802 |

The results of the EFA analysis of the dependent variable show that the observed variables of the voluntary tax compliance variable are all satisfied (factor loading coefficient is greater than 0.5). Therefore, the 06 (six) observed variables of the dependent variable (VTC) remain the same.

*5.3. Linear Regression Analysis*

5.3.1. Pearson's Correlation Test

The Pearson correlation test tests the linear relationship between independent and dependent variables. The test results in Table 5 show that the independent variables PN, AT, SN, KT, FT, and ST with the dependent variable (VTC) all have Sig values less than 5% ($p < 0.01$), so these independent variables are correlated with the dependent variable and will be included in the model to explain the dependent variable.

**Table 5.** Correlations.

| Variables | PN | AT | SN | KT | FT | ST | VTC |
|---|---|---|---|---|---|---|---|
| PN | 1 | | | | | | |
| AT | 0.020 | 1 | | | | | |
| SN | 0.111 * | −0.001 | 1 | | | | |
| KT | 0.122 * | 0.030 | −0.048 | 1 | | | |
| FT | 0.090 | 0.011 | −0.025 | 0.310 ** | 1 | | |
| ST | 0.074 | −0.112 * | 0.059 | 0.079 | 0.061 | 1 | |
| VTC | 0.279 ** | 0.449 ** | 0.213 ** | 0.452 ** | 0.326 ** | 0.150 ** | 1 |

\*. Correlation is significant at the 0.05 level (2-tailed). \*\*. Correlation is significant at the 0.01 level (2-tailed).

5.3.2. Test the Research Hypotheses

This study uses linear regression analysis to test the relationship between the independent and dependent variables, testing the research hypotheses. The regression analysis results in Table 6 show that the adjusted R square is 53.3%, which means that the independent variables explain 53.3% of the variation of the dependent variable. Furthermore, the ANOVA test has a Sig value of less than 5% ($p < 0.01$), so the model is statistically significant and has at least one independent variable affecting the dependent variable. In addition, the demographic variables (age, gender, education, and occupation) as control variables in the model do not influence the dependent variable ($p > 0.1$). This result confirms that the variation of the dependent variable explained by the regression model is mainly caused by the independent variables proposed in the model.

The results in Table 6 continue to show that the independent variables all have Sig values of less than 5% ($p < 0.01$), so the regression model is statistically significant, suitable for the data set, and usable, that is, the independent variables PN, AT, SN, KT, FT, and ST affect the dependent variable (VTC). Furthermore, variance inflation factors (VIF) of the independent variables are all less than 2.20, so multicollinearity is not violated (Nguyen 2011).

Thus, the results of linear regression analysis show that the research hypotheses H1, H2, H3, H4, H5, and H6 are accepted at the 1% statistical significance level. The normalized regression coefficients of the independent variables found were statistically significant, namely PN (β = 0.161), AT (β = 0.444), SN (β = 0.209), KT (β = 0.348), FT (β = 0.193), and ST (β = 0.128).

**Table 6.** Linear regression analysis results.

| Model | | Unstandardized Coefficients | | Standardized Coefficients | t | Sig. | Collinearity Statistics | |
|---|---|---|---|---|---|---|---|---|
| | | B | Std. Error | Beta | | | Tolerance | VIF |
| 1 | (Constant) | −0.386 | 0.306 | | −1.264 | 0.207 | | |
| | PN | 0.124 | 0.030 | 0.161 | 4.117 | 0.000 | 0.905 | 1.105 |
| | AT | 0.248 | 0.021 | 0.444 | 11.745 | 0.000 | 0.968 | 1.033 |
| | SN | 0.151 | 0.027 | 0.209 | 5.511 | 0.000 | 0.959 | 1.042 |
| | FT | 0.189 | 0.039 | 0.193 | 4.908 | 0.000 | 0.897 | 1.114 |
| | KT | 0.362 | 0.041 | 0.348 | 8.749 | 0.000 | 0.873 | 1.146 |
| | ST | 0.130 | 0.039 | 0.128 | 3.383 | 0.001 | 0.960 | 1.041 |
| | Age | −0.056 | 0.039 | −0.056 | −1.421 | 0.156 | 0.902 | 1.108 |
| | Gender | −0.031 | 0.020 | −0.060 | −1.579 | 0.115 | 0.958 | 1.044 |
| | Education | −0.010 | 0.035 | −0.011 | −0.294 | 0.769 | 0.992 | 1.008 |
| | Occupation | −0.020 | 0.018 | −0.043 | −1.139 | 0.255 | 0.982 | 1.018 |
| Adjusted R Square | | | | | | | | 0.533 |
| Durbin-Watson | | | | | | | | 1.873 |
| Sig value of the ANOVA test | | | | | | | | 0.000 |

## 6. Discussion

The regression analysis results in Table 6 confirm that all six factors included in the research model affect voluntary tax compliance behavior. Of which, the possibility of tax inspection and audit has the strongest impact, followed by tax knowledge, social norms, perception of the tax system's fairness, personal norms, and finally, the quality of the tax services has the weakest effect on the voluntary tax compliance behavior of SMEs. The findings of this study are interpreted as follows:

### 6.1. The Possibility of Tax Inspection and Audit

The regression results confirm that the possibility of tax inspection and audit positively affects taxpayers' voluntary tax compliance behavior. In other words, the probability of being audited positively affects the tax compliance behavior of taxpayers, and this finding is similar to the research results of Kirchler (2007) and Inasius (2018). This implies that tax audits can play an important role in enhancing voluntary compliance in self-assessment systems. The rate and thoroughness of audits can encourage taxpayers to be more cautious in completing tax returns, reporting all income, and claiming accurate deductions as necessary to determine their tax liability. Thorough inspection and examination will detect violations if taxpayers do not comply with tax law provisions. As a result, taxpayers are reluctant to receive tax inspections and audits (e.g., Muehlbacher and Kirchler 2010; Kastlunger et al. 2013), so they are more cautious in tax compliance. However, if they violate the tax law, they will most likely be detected and severely handled by the tax authorities, possibly even criminally. Although this may increase tax costs, large penalties for tax non-compliance will pose a high risk to business operations. It is because awareness of severe penalties and possible adverse consequences of non-compliance will lead to better compliance by taxpayers, thereby increasing voluntary tax compliance.

### 6.2. Tax Knowledge

The regression results show that tax knowledge positively affects taxpayers' voluntary tax compliance behavior. As taxpayers' tax knowledge increases, taxpayers' ability to comply with taxes increases. This result is consistent with the assertions of some previous studies (e.g., Eriksen and Fallan 1996; Lewis 1982; Inasius 2015). This finding explains that if taxpayers are knowledgeable about tax law provisions (tax declaration and payment, tax rights and obligations, applicable tax rates, etc.), they will comply with tax well. In the current context in Vietnam, it can be seen that the system of tax regulations and policies

often changes, with many unclear and complicated points; even the implementation of policies at agencies' taxes has not yet been agreed upon. This leads to taxpayers being more susceptible to violations because they do not understand all of the tax policies and procedures changes. On the other hand, the instructions for the implementation of tax regulations have not been detailed and clear, and taxpayers have not been widely communicated to them so that they are not violated. There is still a situation where explaining and guiding tax policies in favor of state management agencies is popular.

In some cases, taxpayers violate tax regulations with severe consequences, even though they do not intend to. Due to the above problems, taxpayers' awareness, consciousness, and attitude towards compliance with the law, including the tax law, is not good and has many limitations. Therefore, to increase the level of voluntary tax compliance, it is necessary to improve taxpayers' attitudes towards taxes.

### 6.3. Social Norms

Social norms continue to be a factor influencing taxpayers' voluntary tax compliance behavior. The findings of this study show that if social norms are raised, taxpayers' ability to comply with tax will increase, then taxpayers will perceive tax violations as wrong is condemned by public opinion, and the tax compliance behavior is right with human morality and is welcomed society. These statements are consistent with the research results of Battiston and Gamba (2016) and Liu (2014). In Vietnam, in recent years, the government and management agencies are paying great attention to the propaganda and dissemination of tax laws in order to raise people's awareness about tax compliance; thereby, taxpayers are more aware and self-conscious in the observance of tax laws. In addition, the propaganda and dissemination of tax legislation will help taxpayers realize the important role of taxes in the country's socio-economic development, ensuring social security for the people. However, people's awareness of compliance with the law, including tax law, is not particularly good. Some people's understanding of tax compliance is still low due to traditional habits, which is the habit of "disobeying the law". Many taxpayers try to find ways to circumvent the law and find loopholes and limitations to commit violations to achieve their own purposes. Therefore, they do not respect the law and indifference and avoidance of the provisions of the law still occur frequently. On the other hand, the Vietnamese cultural value system is being significantly affected by the transition to a market economy, so some taxpayers have degraded moral behavior and value deviation. Therefore, raising the awareness of voluntary tax compliance of the whole society is one of the challenges for the government, policymakers, and tax authorities.

### 6.4. Perception of the Tax System's Fairness

The regression results show that the perception of the tax system's fairness is a factor that positively affects the voluntary tax compliance behavior of taxpayers. This result confirms that a fair tax system plays an important role in determining tax compliance, which is completely consistent with the statements of some previous studies (e.g., Amina and Saniya 2015; Inasius 2018; Taing and Chang 2020). This finding implies that the perception of fairness in tax compliance is one of the great psychological impacts on corporate compliance. If the tax system is perceived as unfair (for example, perceived unfair government spending, perceived tax-related injustice, etc.), then taxpayers are more inclined to evade tax obligations. Therefore, the transparency of expenditures for welfare, anti-corruption, and other social issues of the government (taxes mainly contribute to public spending) will create a sense of fairness in the perception of taxpayers, through which they feel that their contributions are properly spent and benefit society as well as themselves, which will encourage taxpayers to comply voluntarily. In a developing country such as in Vietnam, voluntary tax compliance is still a big challenge for the government and policymakers. In fact, in recent years, the government has had many mechanisms, policies, and solutions to raise people's tax awareness, thereby increasing voluntary tax compliance. However, many taxpayers do not pay much attention to social welfare issues,

and they only care about the benefits obtained by individuals; even quite a few people favor tax avoidance. In addition, some officials in the tax authorities still treat taxpayers with an unfriendly attitude, lack cooperation, and cause difficulties for taxpayers; even though the situation still occurs negatively, this has affected the tax system's fairness.

*6.5. Personal Norms*

The regression results also show that personal norms are a factor that positively affects the voluntary tax compliance behavior of taxpayers. Many studies have confirmed that tax ethics are an important factor in reducing tax evasion (e.g., Wenzel 2004; Alm and Torgler 2006). This study's findings imply that taxpayers with good moral norms are more likely to comply with tax obligations. On the contrary, if their moral norms decline, it is likely that deviant behaviors, such as tax avoidance and tax evasion, will appear. Although, research results show that many taxpayers polled believe that they should honestly declare their income on their tax returns and support voluntary tax compliance. However, in Vietnam today, there are still many taxpayers who support tax avoidance; they support implementing methods that take advantage of loopholes in tax authorities' lax management to reduce the amount of tax payable. Some taxpayers even consider tax avoidance a success and a criterion to prove whether they are good or not or their creativity. Therefore, they do not seem to consider tax compliance as ethical behavior but only consider it a mandatory action according to the provisions of the law. Additionally, many taxpayers are putting personal and business interests first instead of social benefits; they choose economic interests over ethical concerns to achieve their goals. Thus, the issue of how to make taxpayers comply voluntarily and without the need to use mandatory measures is a challenge for the government and policymakers.

*6.6. Tax Service Quality*

The regression results also show that tax service quality positively influences taxpayers' voluntary tax compliance behavior. This finding confirms that improving the tax service quality will create a premise for taxpayers to comply voluntarily, which has also been confirmed in the studies of Obid and Bojuwon (2014) and Hidayat et al. (2014). This study explains that taxpayers who are SMEs have more difficulty understanding tax policy, which can significantly impact their tax compliance behavior. Moreover, tax service is a public administrative service associated with orders and regulations of the state, so the lack of information and instructions associated with the tax service environment will lead to a lack of understanding among taxpayers to comply with tax laws and not create trust and cooperation. Therefore, when tax authorities create trust in taxpayers through the service environment, it will incentivize them to increase voluntary tax compliance. Once taxpayers are satisfied with the tax services of the tax authorities, they will feel happy to follow the recommendations of the tax authorities or tax regulations of the state, thereby increasing their voluntary tax compliance without the need to use mandatory measures to comply with tax through sanctioning tax violations. The reality of the past shows that Vietnam's economy has made great progress, so the quality of public services has also improved significantly compared to the past, including the quality of public services of tax authorities. Some tax officials have an unfriendly attitude and are not open-minded, interested, and empathetic with taxpayers in solving work. Along with that, the information technology system infrastructure has not been able to keep up with the speed of the tax administration reform and modernization requirements, with the change of the tax legal policy system; the level of integration and automation is not high; the information technology human resources of the tax industry are relatively thin, and the quality is not guaranteed. This leads to the tax authorities' low quality of public services, which has not created taxpayers' trust, cooperation, and satisfaction, affecting taxpayers' voluntary tax compliance behavior.

## 7. Conclusions and Recommendations

Taxes are considered the main source of budget revenue of a country, an important tool for the government to regulate the macroeconomy, promote investment, redistribute wealth and income in society, etc. Therefore, the issue of tax compliance is of great interest to policymakers in countries around the world, especially in developing countries. One of the main problems plaguing policymakers in developing countries is coming up with solutions to encourage higher levels of tax compliance. Stemming from issues of particular interest to the government and policymakers in Vietnam, this study examines non-economic factors affecting the voluntary compliance behavior of SMEs in Vietnam. The data used in the study were collected through a survey of managers (directors, deputy directors, chief accountants) and tax accountants at SMEs in the provinces and cities in Vietnam and processed by statistical software SPSS 20. The results of linear regression analysis confirm that:

The possibility of tax inspection and audit strongly influences taxpayers' voluntary tax compliance behavior. Taxpayers all said that they are aware of the heavy fines as well as the adverse consequences that may be encountered if the tax authorities detect that they do not comply with the tax through the inspection. Therefore, tax authorities and related agencies need to strengthen tax inspection and examination in terms of both quantity and quality and, at the same time, increase the level of penalties if taxpayers are found to have violated tax laws in order to improve voluntary tax compliance of taxpayers.

The tax knowledge of many taxpayers is still limited, leading to their awareness and attitude toward tax compliance being insufficient. Therefore, to limit violations and promote voluntary tax compliance, tax authorities and related agencies need to have solutions to improve taxpayers' tax knowledge by simplifying administrative procedures and strengthening training, propaganda, and dissemination of tax policies, thereby improving the compliance attitude of taxpayers.

Currently, the social norms of many taxpayers support tax avoidance; they have not considered tax compliance behavior as ethical behavior but only consider it a mandatory action according to the provisions of the law. Tax compliance has not been considered a standard of behavior of taxpayers in a social community, so the social criticism of tax non-compliance will be the most severe punishment. Therefore, policymakers and tax authorities need to pay attention to improving the tax culture in implementing tax management strategies, thereby creating a culture of tax compliance for people and businesses.

The tax system's fairness is being affected because some officials in the tax authorities treat taxpayers with an unfriendly attitude, and a lack of close coordination causes difficulties for taxpayers, even negative situations occur. Therefore, policymakers and tax administration agencies need to continue to promote tax system reform and administrative procedure reform and strengthen the discipline of tax administrators in accessing the tax laws of taxpayers, avoiding unnecessary barriers. On the other hand, the government needs to strictly control and transparently spend public spending so that taxpayers feel the tax system is fair to them, thereby encouraging them to voluntarily comply with tax laws.

The personal norms of many taxpayers are not good, and they have not considered tax compliance behavior as ethical behavior but only consider it a mandatory action according to the provisions of the law. As a result, many taxpayers are putting their personal or business interests first instead of the good of society. Therefore, policymakers and tax administration agencies should continue to propagate to the people to understand the rights and obligations of tax payment of citizens, the role of taxes in socio-economic development, and protect the legitimate interests of each individual. In addition, these agencies need to develop soft skills training programs for learners who have a sense of compliance with the law, in which tax compliance is essential and should be better conducted. In the long term, this will improve people's knowledge and raise the tax compliance consciousness of the whole society.

In recent years, the tax service quality in Vietnam has improved significantly; however, the quality is still not high, creating trust, cooperation, and the satisfaction of taxpayers,

affecting the behavior of voluntary tax compliance by taxpayers. Therefore, to increase voluntary tax compliance, policymakers and tax authorities need to focus on reforming tax administrative procedures and building and developing human resources in the tax industry to ensure professionalism to promptly solve problems, create comfortable psychology for taxpayers when working in person or an electronic environment, thereby building trust and creating a good image for tax authorities. In addition, tax authorities need to invest in equipment and facilities for the most effective electronic tax declaration and payment. In addition, tax authorities need to regularly review to have solutions to remove difficulties and obstacles of taxpayers in implementing tax policies to create favorable conditions for taxpayers to comply with tax obligations.

Although the findings of this study have achieved certain results aimed at helping policymakers and tax authorities incorporate measures to address these difficulties in system development or decision-making tax policies that promote voluntary tax compliance by taxpayers. However, this study still has some limitations, and further studies should consider expansion that will provide more valuable results, such as consider increasing the number of non-economic factors such as gender, age, ownership structure, etc.; expanding the sample size to represent all taxpayers; focus on each specific type of enterprise, such as state-owned enterprises, private enterprises, foreign-invested enterprises, etc.

**Funding:** This research received no external funding.

**Institutional Review Board Statement:** Not applicable.

**Informed Consent Statement:** Not applicable.

**Data Availability Statement:** Not applicable.

**Conflicts of Interest:** The author declares no conflict of interest.

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
