# Peer review of "The Impact of Non-Economic Factors on Voluntary Tax Compliance Behavior: A Case Study of Small and Medium Enterprises in Vietnam"

_economies, doi:10.3390/economies10080179_

Round 1
Reviewer 1 Report
Drop section 2.1 , the theory of planned behavior. The seminal theoretical work on tax compliance is Allingham and Sandmo (1972) that could be used as the starting point of the literature review. However, the theoretical underpinnings can be traced further back in Becker (1968) which is missing.
Another key omission in the literature review is the lack of any mention of “tax morale” as another term used in the literature to describe voluntary tax compliance. In fact, the author(s) may want to update the definition of voluntary tax compliance referenced in Kirchler (2008) to the standard definition of tax morale in Luttmer and Singhal (2014).
Section 5 – Research Results
It is unclear what we learn from sections 5.1 and 5.2. The author(s) suggest that the data are suitable for further analysis following assessment of correlation coefficients and exploratory data analysis. For brevity, this information could be summarized in a footnote.
The naming convention followed PN, AT, … makes results hard to read and easily interpret. It is recommended that the authors provide the full name of the variable or at least an abbreviation that hints towards the relevant hypothesis.
More importantly, the authors should not assess the importance of each variable based on the estimated magnitude of the coefficient. These are a function of the variable scale and are not directly comparable. Instead, other data mining techniques may be used such as LASSO regression, a method used in the tax morale literature to assess the importance of each variable and guide model selection (Koumpias, Leonardo and Martinez-Vazquez).
References
Becker, G. S. (1968). Crime and punishment: An economic approach. In The economic dimensions of crime (pp. 13-68). Palgrave Macmillan, London.
Luttmer, E. F., & Singhal, M. (2014). Tax morale. Journal of economic perspectives, 28(4), 149-68.
Koumpias, A. M., Leonardo, G., & Martinez-Vazquez, J. (2021), “Trust in Government Institutions and Tax Morale”, FinanzArchiv / Public Finance Analysis, 24, 1-24. https://doi.org/10.1628/fa-2021-0006
Reviewer 2 Report
Similar studies have been done many times in other countries. The author can emphasize the difference.
Reviewer 3 Report
Please refer to the attachment.

Reviewer 4 Report
The paper addresses a topic of great interest and actuality. This study examines non-economic factors affecting the voluntary tax compliance behavior of small and medium enterprises (SMEs) in Vietnan.
The article provides an interesting analysis of the subject, but it is important that author completes the literature review and updates it. It would also be interesting to include summary tables on the topics addressed in the review.
Round 2
Reviewer 3 Report
Please see the attachment.
